# Central Pattern Generator with Defined Pulse Signals for Compliant-Resistant Control of Biped Robots

**DOI:** 10.3390/biomimetics8010100

**Published:** 2023-03-02

**Authors:** Zihan Xu, Qin Fang, Chengju Liu, Qijun Chen

**Affiliations:** 1Robot and Artificial Intelligence Lab (RAIL), College of Electronic and Information Engineering, Tongji University, No. 4800 Cao’an Road, Shanghai 201804, China; 2Tongji Artificial Intelligence (Suzhou) Research Institute, Suzhou 215000, China

**Keywords:** balance control, biped robot, trajectory generator

## Abstract

For biped robots, the ability to maintain balance under external forces is an essential requirement. Inspired by human beings’ behaviors to resist external forces, a compliant-resistant balance-control method is proposed to keep the biped robot balance subjected to an external force. A model-free trajectory generator is designed based on the central pattern generator (CPG) to generate compliant-resistant human-like behavior. The CPG pattern generator generates the desired pulse signal utilizing Matsuoka’s CPG. The signal modulator applies the defined signal to the robot’s center of mass (CoM) to generate the workspace trajectory when standing on double feet. Moreover, when standing on single foot, the output signal of the CPG will directly act on the hip joint of the robot to generate the joint space trajectory. Furthermore, the motion engine calculates the workspace trajectory into joint sequence values. The proposed control strategy can generate defined pulse signals to realize compliant-resistant balance control for biped robots. The control strategy proposed in this paper is verified in the NAO simulation environment.

## 1. Introduction

Humanoid robots appear as helpers who assist or replace human beings in accomplishing some tedious and dangerous work because of their higher flexibility and adaptability. Humanoid robots are bionic robots with a human-like torso, arms, legs, etc. Compared with quadruped and wheeled robots, biped robots have advantages of good ground adaptivity, tremendous obstacle avoidance ability, and unlimited potential applications in many fields, such as military, rescue, and space exploration [1]. It makes sense for biped robots to leave the laboratories and traverse the real world. Thus compliance and more efficient locomotion are necessary [2]. The research on biped robots has been thoroughly carried out since the appearance of the world’s first humanoid robot, WABOT-1, in 1973 [3]. After half a century of development, biped robots have made significant progress in hardware, including mechanical structure, materials, sensor accuracy, etc., and algorithms on control systems, balance, and gait generation [4,5,6]. Biped robots have attracted a lot of attention, with a vast amount of research focusing on biped robots’ balance and walking control.

Maintaining the balance of biped robots is challenging since biped robots consist of links and joints, resulting in multiple joint degrees of freedom (DoFs). Kajita et al. [7] proposed a method using a preview control to balance the biped robot, and the method was validated on the biped robot HRP-2 prototype (HRP-2P) [8]. One control method [9] was raised based on the biped robot Cassie and the robot has the ability to keep balance in complex environments. Ref. [10] proposed a dynamic torso compliance method for biped robots to maintain balance when subjected to unknown external forces. Humanoid robots accepting external forces without falling down is the essential requirement in robot motion control, and it is also a difficult and complicated problem that needs to be handled property [11]. The position-controlled humanoid robotics has the property of resistance inherently [12], since actuators fixed on the robots are under control of PID controller. Generally, the humanoid robot is regarded as a rigid body in dynamic analysis [13]. It means that the structure of the rigid body does not change when external forces act on it. When the robot is subjected to an external force, the joints will rotate accordingly, even if the position controller makes efforts to maintain the position of the joints. Fortunately, the rotation range of the robot joints is restrained under the control of the position controller. Afterward, the angle position errors between the expected output and the measurement are calculated in view of the feedback of the position controller, the PID controller in general. The PID controller integrates the derivation and soon reaches saturation, making the robot oscillate suddenly in the opposite direction if external forces acting on the robot suddenly withdraw.

These problems make humanoids less suited for collaborating with humans and working in their environments. In such situations, a critical issue that the humanoid robot needs to face is dealing with unknown external disturbances. Compliant control has been proven that it allows biped robots to remain stable in tasks involving interactions with complex environments [14]. Various novel motion-control methods have been proposed [15,16,17,18,19,20]. These papers realize the compliant control by passive compliance actuators [15,17,21]. A torque-controlled robot can distribute all task force/torques to the end effectors and make direct force control [16,18,22]. Elastic actuators with compliant structures also have problems in trajectory tracking, stored energy releasing, and so on [19,23,24]. Ko et al. [18] combined both electronic and hydrostatic actuators and realized the compliant locomotion with back drivability and high control bandwidth. Li et al. [20] proposed a compliance control method based on a viscoelastic model for a position-controlled robot. Liu et al. [25] raised a locomotion method based on CPG generators for humanoid robots.

Biological studies have found that the Central Pattern Generators (CPGs) located at the spinal cords can produce coordinated oscillatory signals without oscillatory inputs and control many rhythmic biological movements, such as breathing, swimming, and walking. Kim et al. [26] proved that the neural system (CPGs with reflexes) and the musculoskeletal system could generate the interaction between rhythms to produce coordinated bipedal walking. Inspired by biological CPGs, the CPG concept has been applied to biped robot locomotion control. Many CPG robot models have been proposed from different perspectives, such as biophysical models, oscillator models, etc. Most of the traditional CPG models can only generate sinusoidal or sinusoidal-like signals, which significantly limits the adaptability of CPG trajectories. However, the trajectory required by humanoid robots cannot be simulated only with sinusoidal or sinusoidal-like signals. A CPG design method was raised based on Fourier series to generate special waves [27]. Matsuoka et al. [28] proposed a model which could generate trajectories of any shape. A CPG topological network is constructed with CPG units assigned to degrees of freedom (DoFs). Simultaneously, complex multi-dimensional coordination signals are generated, and joint motion is directly controlled to realize motion control. Generally, Humanoid robots have numerous DoFs. Thus the CPG network used in the traditional method will be very complicated and difficult to be optimized.

Inspired by the resistant compliance robot locomotion control strategy proposed by Huang et al. [14], we design a compliant-resistant controller based on a CPG trajectory generator. Different from other model-based biped robot control algorithms, our proposed strategy is model-free, utilizing the CPG unit to yield signals. In this paper, we utilize Matsuoka’ CPG model as the CPG pattern generator to yield our required pulse signals. Considering the redundancy of the CPG units mentioned above for biped robot joint control, when the biped robot stands on double feet, we propose to act the desired signals on the center of mass (CoM) of the biped robot; When the biped robot stands on a single foot, put the desired signals on the hip joint of the support leg. Different from the “CPG-joint control method” [29,30], the CoM trajectory (or workspace trajectory) is mapped into robot joint space through the biped robot kinematics and dynamics analysis. In this paper, we propose a CPG-based method of a compliant-resistant balance controller for the biped robot balance control. A desired signal will be provided by the CPG unit to mimic posture changes of the human body when resisting an external force and buffer the impact of the external force acting on the robots in real-time.

The compliant locomotion control strategy for biped robots is verified on the humanoid robot Nao in the simulation environment. In our paper, we use the deviation between the measured ZMP value and the reference ZMP value to judge whether the biped robot is contacted by an external force. However, the Nao robot is equipped with simple force sensors (Fsr). These simple Fsrs cannot provide accurate value and direction of contact forces directly. In order to handle the poor performance of the Fsrs, we present an online compliant-resistant controller with Gravity Projection Observer (GPO). The controller achieves the realization of disturbance absorption, keeping the humanoid balance. Furthermore, our algorithm maintains the robustness of the humanoid contact with environments with the proposed controller. Following the natural human response, modulating the CPG signals to maintain the biped robot balance when the robot is subjected to a small external force in the situation of standing on a single foot or double feet. We design a compliant-resistant balance controller for the biped robot. These are the main contributions of this paper.
•A CPG pattern generator is proposed based on Matsuoka’s CPG unit. It generates the desired pulse signals required to achieve the compliant-resistant balance control for the biped robots;•A signal modulator is raised to map the output signals of the CPG model to the workspace trajectory (when standing on double feet) or joint space trajectory (when standing on a single foot) of the biped robots;•A novel method is proposed to design Matsuoka’s CPG to generate the pulse signals required;•The proposed control strategy is validated in the biped robot NAO simulation environments.

The remainder of this paper is structured as follows: In Section 2, we introduce the architecture of our proposed algorithm. In Section 3, the CPG-based trajectory generator will be presented. Section 4 describes the framework of our proposed controller. Section 5 provides the results of simulation experiments. Finally, we conclude the study and suggest future works in Section 6.

## 2. Control System Architecture Based on CPG

In this paper, the control system architecture, shown in Figure 1, consists of a CPG pattern generator, a signal modulator, and a motion engine. The CPG pattern generator (composed of a Matsuoka’s CPG module) generates signals output based on the designed pose signals inspired by human behavior against external forces. The signal modulation stage will be divided into two cases. When the biped robot stands on double feet, the signal outputs, generated by the oscillators acting on the *x*-direction and *y*-direction, is applied to the center of mass (CoM) of the robot. The CoM trajectories of the biped robot can be generated based on the combined output of the oscillators during the double support phase (DS). Then, the predesigned workspace trajectories can be modulated by the generated CoM. That is, the workspace trajectories will be designed as the combined output signals of the CPGs. When the robot stands on a single foot, the output signals of the CPGs directly act on the hip joints of the biped robot, the pitch joint, and the roll joint. The hip joints trajectories are generated as the function of the combined outputs of the above-mentioned CPGs during the single support phase (SS). In the case of the biped robot standing on double feet, use the motion engine to calculate the joint angle sequence values of the robot through inverse kinematics calculation based on the workspace trajectories generated by the signal modulator. Furthermore, the output value feed of the contact force estimation is used as the sign to adjust the pose of the biped robot to keep balance.

## 3. CPG-Based Trajectory Generation and Optimization

### 3.1. CPG Model

Many research works designed rhythm signals based on Matsuoka’s CPG model [31,32]. However, Matsuoka’s CPG model has the ability to generate signals of any shape. One CPG unit consists of two mutually inhibiting neurons, corresponding to a flexor neuron and an extensor neuron in human beings. In this paper, Matsuoka’s CPG model mentioned above can be formulated as: (1)$$\left\{\begin{array}{cc} T_{u}{\dot{u}}_{i} & =-{\dot{u}}_{i}-w_{fe}y_{j}-bv_{i}+s_{0}+feed_{i}\\ T_{v}{\dot{v}}_{i} & =y_{i}-v_{i}\\ y_{i} & =\max(u_{i},0)\\ y & =y_{2}-y_{1} \end{array}\right.$$
where subscript *i* represents 2 or 1, denoting an extensor neuron and a flexor neuron, respectively. Variable $u_{i}$ represents the inner state, and $v_{i}$ is a variable to indicate the degree of self-inhibition effect. Variable $y_{i}$ is a representation of the output of the extensor or flexor neuron, and $s_{0}$ is the external input. $T_{u}$ and $T_{v}$ are the rising time and adaptation time constants. Parameter $w_{fe}$ represents the connecting weight between flexor and extensor neurons. The constant representing the degree of self-inhibition is parameterized as *b*. *y* denotes the output of the CPG model.

In this paper, we define one CPG unit consisting of two Matsuoka’s neurons, which can generate pre-designed signals of any shape. We utilize the parameter feed as the feedback signal. We propose to ignore the third equation of (Equation 1) when generating a non-rhythmic signal. On the contrary, it will be retained to calculate the output of the CPG model.

### 3.2. Phase Portrait

In a dynamical system, if the forward trajectory of any point on a track is in a small enough area, or if the whole trajectory stays in one area as a whole, then the state of the trajectory is stable. Matsuoka’s CPG is a 2N-dimensional dynamic system. With different parameter settings, a CPG model has the ability to generate stable and unstable signals. There is a flexor neuron and an extensor neuron in one Matsuoka neuron. During the trajectory optimization stage with $feed_{1}$ and $feed_{2}$ disabled, four state variables are $u_{1}$, $v_{1}$, $u_{2}$, $v_{2}$. Figure 2 shows the output driven by parameters [$T_{u}$, $T_{v}$, $s_{0}$, *b*, $w_{fe}$]. Figure 2a shows the time domain signal generated by the CPG model. Figure 2b plots phase portraits, which are in the $u_{1}-v_{1}$ plane, the $u_{1}-u_{2}$ plane, the $u_{1}-u_{2}$ plane, the $v_{1}-v_{2}$ plane, and the $u_{2}-v_{1}$ plane, respectively. Figure 2c plots the difference between the four state variables, respectively. These plots of the geometric trajectories in the state space reveal that an attractor is present for the chosen parameter values. Thus the generated signal can decay to zero over time and be stable.

### 3.3. Defined Pulse Signal

The CPG model generally generates periodic rhythm signals that can realize the robot’s periodic movements. For biped robots, walking, accomplished by periodic alternating back-and-forth motions of the legs, is a common periodic movement. In this work, we design a CPG model with the characteristics of pulse signal output. After the output pulse signal is modulated, it is bound to the robot’s workspace motion controller when the biped robot stands on double feet and joint space motion controller when on a single foot, respectively, achieving the result of compliant-resistant control. Compliant and resistant motion is the movement behavior of human beings against external forces. Inspired by this, We assume that a biped robot will behave in compliant-resistant motion like human beings when subjected to external forces. Taking the force in the positive direction of the *y*-axis as an example, the same motion direction as the force direction is called the compliance process, and the opposite motion direction is called the resistance process, and vice versa. The character of compliance and resistance represented in the time domain can be simplified as a half-period sinusoidal signal or a peak of a $\mathrm{sinc}\left(t\right)$ signal. After undistorted scaling and translation, these signals have similar characteristics: they are excited from zero and decay to zero. Only the first rising stage of the function is the compliance stage, and the process of the function signal decaying from the peak value is the resistance stage.

In this paper, we adopt Matsuoka’s CPG as the signal generator. This CPG model is driven by five parameters, [$T_{u}$, $T_{v}$, $s_{0}$, *b*, $w_{fe}$], which can be regarded as a parameter vector. Our design goal is to find a suitable parameter solution vector so that the output signal of the CPG model has the characteristics of compliance and resistance. The process of finding a parametric solution can be divided into the following steps:•Randomly generate an initial set of parameters;•Numerically integrate the CPG model under each parameter definition to obtain the time domain signal;•Evaluate the similarity of the obtained signal to the given signal characteristics, expressed in fitness;•Update parameters based on similarity;•Repeat the above process to acquire the parameter solution with the most prominent similarity.

The Particle Swarm Optimization (PSO) optimization algorithm is suitable for the above parameter optimization process. Each particle represents a parameter solution vector in this algorithm, and a current position vector and a current velocity vector define the state of the particle. The particle swarm updates the velocity vector according to the optimal solution vector of all particles and the optimal solution vector that each particle has reached and then updates the position vector to realize the optimization of parameters. The particle update process can be calculated in parallel to obtain a higher iteration speed. In the above optimization process, defining a fitness function that can represent compliant-resistant features is the most important. The higher fitness means that the CPG model driven by the current parameters has better compliant-resistant characteristics. The PSO optimization algorithm will iteratively calculate the particles with the best fitness according to the fitness function and then obtain the defined optimal CPG parameters.

In order to obtain a signal with compliant-resistant characteristics, we believe that the time-domain signal integrated by the CPG model should have the following characteristics:•The signal can complete the compliance and resistance action process within a given time range and remain stable;•The signal can eventually decay to zero;•The signal should have a particularly high energy, which is specifically reflected in the fact that the output amplitude of the original signal cannot be too small. Otherwise, the control signal will not reflect the characteristics of compliance;•The time integral of the absolute value of the signal cannot be too large. The CPG output signal corresponds to the control quantity of the robot controller, and the integral result corresponds to the cumulative value of the control quantity output by the controller. It’s desired that the controller can use as little control quantity as possible to achieve the control result.

The time-domain signal obtained this way is excited from zero and decays after reaching the peak value. It is different from the rhythmic signal generated by general CPG. The shape of the signal is similar to a pulse with a specific amplitude and width. This is why we call the CPG output with the compliant-resistant characteristics pulse signal.

### 3.4. CPG Pattern Optimization

The fitness function is critical in guiding individuals to a good solution. Unlike the general optimization process of nonlinear problems, there is no intuitive nonlinear equation to represent the generated CPG pulse signal, which means that the signal generated according to some parameter solutions is what we want, which means the fitness function can be used to calculate the fitness of the signal. However, the signal generated by other parameters is not and its fitness is unreachable. Set the fitness of the feasible solution to 0-1, and a fitness of 1 means that it completely fits the signal characteristics. Set the fitness of the infeasible solution to a negative number to distinguish the feasible solution from the infeasible solution. It is also convenient for the particle swarm optimization algorithm to calculate the average result of all particles. The particles corresponding to infeasible solutions can be eliminated according to the fitness sign. First, consider giving parameter constraints to exclude impossible solutions, and then define the solution equation corresponding to each feature according to the compliant-resistant signal characteristics, and the fitness function will be determined by the calculated value of each characteristic equation.

The infeasible solution is limited by six constraints, defined as follows:•Denominator is zero;•Wrong direction;•Overshoot constraint;•Derivative constraint;•Unstable signal;•Not cross zero.

In Equation (Equation 1), $T_{u}$ and $T_{v}$ are the denominators of the equation in the numerical integration calculation and cannot be set to zero. When initializing particles, the particles generated by random numbers should ensure that $T_{u}$ and $T_{v}$ are not zero. During the particle update process, if $T_{u}$ and $T_{v}$ are zero, set the fitness of the particle to a negative number. The output direction of the signal in the compliance phase should be consistent with the direction of the force. Assuming the force direction is positive, the signal should also be output in the positive direction during the compliance phase. The peak of the signal means the farthest position the robot reaches when performing a compliant motion. According to our experience, some parameter solutions will cause the peak of the signal to reach infinite or tend to zero, and such values will bring some unnecessary numerical calculation troubles. In our design, when the peak value of the output signal exceeds 100 or is less than 0.01, the set of parameter solutions is considered an infeasible solution. In fact, this limitation does not require much consideration, and it is just for the convenience of excluding some potential dangers in the value. The output signals of the CPG model need to be modulated into control signals which can be accepted by the biped robot. Due to the limitation of the driving ability of the robot actuator, the absolute value of the derivative of the output signal should be controlled within a reasonable range. In order to unify the comparison standard, we normalize the signal and limit the peak value to 1. In fact, in the constraints of the peak value listed above, it is ensured that the signal here can be normalized.

We use the absolute value of the maximum derivative of the $\sin(\frac{2}{3}\pi t)$ signal as the reference benchmark slope, and test the optimization results at 1 times the slope, 2 times the slope, 3 times the slope, 4 times the slope and 5 times the slope, as Figure 3 shows. The symbol of × denotes times. The signal decays slowly under the 1× and 2× slope constraints, and the signal changes a little bit fast under the 4× and 5× slope constraints. Considering that the 3× constraint is enough, we choose 3× the slope as the constraint. The trajectory of state variables should remain stable and be confined to a limited region. We set this area as a circle with a radius of 1, i.e, the deviation between any state variables must not exceed 1. Theoretically, it is not necessary for the signal to cross the zero point. In order to artificially strengthen the resistance characteristics of the pulse signal, we make the signal must have zero overshoot. When the output signal does not satisfy the above six constraints, fitness is set to a negative number.

The cost time of a signal is the time until the signal reaches the termination condition. When the pulse signal degenerates approximately to a direct-current (DC) signal, the signal is considered to have reached the termination condition. We utilize the method of time sliding window to record the maximum value of the absolute value of the signal within 1 s and judge whether it is less than the reference value. Or it is considered that the DC signal is detected if the caculated difference between the maximum value and minimum value is within the limit. Record this time as terminal. The reference value and the limit is set to 0.01. When the signal reaches the maximum integration time of 5 s, the cost time is recorded as 5 s. The maximum fitness is obtained when the signal cut-off time is around 1.5 s. The fitness function about the cut-off time is expressed as: (2)$$f_{1}\left(x\right)=\left\{\begin{array}{cc} \frac{1}{\sqrt{2\pi }\times 0.6}\exp\left(-\frac{{\left(x-1.5\right)}^{2}}{2\times 0.6^{2}}\right) & x⩾1.5\\ \frac{1}{\sqrt{2\pi }\times 0.6}\exp\left(-\frac{{\left(x-1.5\right)}^{2}}{2\times 0.6^{2}}\right) & x<1.5 \end{array}\right.$$

The difference between the output of the signal at the end and the start time is regarded as a terminal error. We expect the signal to start at zero and eventually converge to zero and back and forth. The fitness function of terminal error is expressed as: (3)$$f_{2}\left(x\right)=e^{-1}\left(\frac{\left|x\right|}{10}\right)$$

The peak value of the signal reaching within the cut-off time represents the maximum energy of the signal, which is called overshoot. Due to the limitation of cut-off time and cumulative control amount in optimization conditions, the optimization result may lead to zero, and the signal can satisfy the above two conditions simultaneously. In order to prevent the optimization result from collapsing into this situation, a certain reward is given to the overshoot of the signal, and the overshoot fitness function is expressed as: (4)$$f_{3}\left(x\right)=\left\{\begin{array}{cc} \frac{1}{\sqrt{2\pi }}\exp\left(-\frac{{\left(x-10\right)}^{2}}{2}\right) & x⩾10\\ \frac{1}{\sqrt{2\pi }}\exp\left(-\frac{{\left(0.2x+12\right)}^{2}}{2}\right) & x<10 \end{array}\right.$$

The magnitude of the signal affects the result of integrating the absolute value of the signal over time. So in calculations, the signal is normalized to have an amplitude of 1. For digitally sampled signals, the integration of the signal over time can be replaced by the accumulation of signal values. For a sin(t) signal, the ratio of the integral over half a period to the integral of a signal with amplitude 1 is $\frac{2}{\pi }$, that is: (5)$$\frac{\int_{0}^{\pi }\sin\left(t\right)dt}{\int_{0}^{\pi }1dt}=\frac{2}{\pi }$$

Suppose the ratio of the accumulated value of the output signal to the accumulated value of the signal with an amplitude of 1 within the cut-off time is greater than $\frac{2}{\pi }$. In that case, the signal does not meet the goal of optimal control. The fitness function of the cumulated control cost is expressed as: (6)$$f_{4}\left(x\right)=\left\{\begin{array}{cc} \frac{2}{\pi } & \frac{x\times T}{T_{c}}>\frac{2}{\pi }\\ 0.49\cos\left(\frac{\pi^{2}}{2}\frac{x\times T}{T_{c}}\right)+0.5 & \mathrm{else} \end{array}\right.$$
where *T* is the sampling period and equal to 0.01, $T_{c}$ is the cut-off time, which means the integration end time.

In summary, the fitness function of the pulse signal is jointly determined by the above fitness functions, expressed as: (7)$$fitness=f_{1}\times \left(0.4f_{2}+0.4f_{3}+0.2f_{4}\right)$$

### 3.5. Signal Correlation Analysis

The optimization function definition mentioned above is tedious and complicated, but it is necessary to obtain the unknown characteristic signal. We have obtained the desired pulse signal according to the proposed time-domain signal performance index. If we want to change the time-domain signal index and obtain another signal waveform, we need to re-correct the fitness function. It is because we cannot intuitively summarize the impact of different parameters on the final model output from the CPG model. Nevertheless, we can still try to model and analyze the obtained pulse signal and express the pulse signal as a combination of some basic elementary functions. Then we can easily define a template signal that meets the performance index. Calculate the cross-correlation between the CPG signal and the template signal generated under different parameters. Then the fitness function can be expressed by the cross-correlation between the signals.

We use the combination of $e\left(t\right)$ and $\sin\left(t\right)$ to realize the periodicity and attenuation characteristics of the pulse signal. Such signals are used as template function: (8)$$f\left(t\right)=Ae\left(-\tau t\right)\sin\left(\omega t\right)$$
where *A* determines the amplitude of the template signal, τ determines the cut-off time of the template signal, and *w* determines the period of the template signal. Although the attenuation function does not have an actual period, the time when the signal crosses the zero point from the positive semi-axis can still be determined according to *w*. When A=2.5386, τ=1.91387, ω=2.50499, the template signal, plotted in the red line, can fit the CPG output well, plotted in black stars, shown in Figure 4. The calculation of signal correlation can be expressed as a convolution operation of two signals. We know that convolution in the time domain is equivalent to multiplying one signal by the conjugate of another signal in the frequency domain, that is: (9)$$s_{sam}\left[n\right]*s_{tem}\left[-n\right]=S_{sam}\left[k\right]\cdot S_{tem}^{*}\left[k\right]$$
where $s_{tem}$ and $s_{sam}$ are the time-domain signal of the template function and the sampled CPG output, respectively, $S_{sam}$ is the Fourier transform corresponding to the sampling signal of the time domain, and $S^{*}$ denotes the complex conjugate of the signal.

To calculate the correlation between the reference signal and the unknown signal sampled from CPG outputs, we transform the signals into the frequency domain by a Fast Fourier Transformation (FFT). We use a highly efficient FFTW3 implementation as the calculation method of FFT [33]. Set the first calculated element of the frequency domain to zero, remove the DC component in the signal, and then transform the signal from the frequency domain to the time domain through Inverse Fast Fourier Transformation (IFFT). Normalize the time-domain signal to a signal with an amplitude of 1. The resulting normalized signal is then transformed into the frequency domain. The signal spectrum at this time can be used as the eigenvector of the signal. Transform the calculation results of the frequency domain to the time domain, and at this time, the maximum value of the time-domain signal is used as the cross-correlation value of the signal. When the two comparison signals are the same signal, it is also called the autocorrelation of the signal. Suppose the cross-correlation value of the signal is closer to the autocorrelation value, and it indicates that the two signals are more similar. Therefore, we use the ratio of cross-correlation and autocorrelation values of the two signals as the calculation item of the fitness function, which can be a good substitute for the Formula (Equation 7). A fitness function defined using the correlation of the sampled signal and the autocorrelation of the reference signal is expressed as: (10)$$fitness=\frac{correlation}{autocorrelation}$$

## 4. Control Framework

In the previous section, we propose the CPG-generating method for defined pulse signals. The oprimized parameters of the CPG model are $T_{u}=3.10412$, $T_{v}=0.424391$, $s_{0}=9.28835$, b=9.31659, $w_{fe}=-2.99902$. To verify the balance performance of the designed CPG pattern generator in response to the unknown extern impact, we bind the CPG output signals at the center of mass and the hip joints of the robot, respectively. Two CPG signals are mapped at the CoM as workspace trajectories. One signal controls the movement of the CoM in the *x*-axis direction, and the other controls the movement in the *y*-axis direction. In this work, the robot stands on the ground, which is usually the case, using the double feed. The modulated CPG output at the CoM can adjust the position of the center of mass according to the magnitude of the contact force, and the effect of the force can be offset utilizing the compliant resistance signal. When the robot stands on the ground on foot, which is a challenging action, the modulated CPG output signals at the hip joints can be directly applied to the joint rotation control. The case where the robot stands by a single leg can be divided into the left and the right. The CPG pattern generator act on the left hip joint if left foot support, and the output of the CPG generator acting on the right hip joint will be disabled, vice versa.

### 4.1. Mapping Signals

We transform the dimensionless signals of CPG outputs into the CoM workspace by multiplying translation gains, $K_{x}$ and $K_{y}$, in the *x*-axis direction and *y*-axis direction, respectively. The function mapping process of the CoM position control is as follows: (11)$$\left\{\begin{array}{cc} c_{x} & =c_{x_{0}}+K_{x}y_{c_{x}}\left(t\right)\\ c_{y} & =c_{y_{0}}+K_{y}y_{c_{y}}\left(t\right) \end{array}\right.$$
where $K_{x}=20$ and $K_{y}=20$. These two parameters are adjustable based on the maximum compliant distance of the CoM. $c_{x_{0}}$ and $c_{y_{0}}$ is the initial position of the CoM in the world frame. In this work, The initial position of the CoM is set to the origin of the world frame. The direction of movement in the CoM work space is consistent with the CPG outputs. The CPG outputs a positive signal makes the CoM move along the positive direction of the *x*-axis or the positive direction of the *y*-axis.

In joint space, we perform a similar operation but need to pay attention to the sign of the CPG output signals. When the contact force from the positive direction of the *x*-axis acts on the robot, it should lean forward to achieve compliant motion. The forward lean is achieved by reducing the angle of the hip pitch joint, so the CPG output signal needs to be reversed. The mapping functions in joint space are as follows: (12)$$\left\{\begin{array}{cc} pitch & ={\mathrm{offset}}_{pitch}-K_{p}y_{pitch}\left(t\right)\\ roll & ={\mathrm{offset}}_{roll}+K_{r}y_{roll}\left(t\right) \end{array}\right.$$
where $K_{p}=0.0873$ and $K_{r}=0.0873$. ${\mathrm{offset}}_{pitch}$ and ${\mathrm{offset}}_{roll}$ is the initial rotation angle to support the stand.

### 4.2. Feed

The CPG generator generates the pulse signal in the initial state as shown in the first row in Table 1 without feeds. The state variables change to a steady state after five seconds, as shown in the second row in Table 1. In the balance control, we choose the steady state as the initial state to ensure that the robot is in a static state. Otherwise, the robot will move itself without any operation intervention, which is unacceptable. The contact force is chosen as the feed as the feedback for the balance control.

We take NAO robot as the experimental platform of the CPG pattern generator. The force sensors of the NAO robot are equipped on the bottom of the feet, four on the left foot and the other four on the right foot. The contact force cannot be measured directly, so we estimate the force by combing the robot configuration, kinematics and inertial sensor data. Based on the point mass model as shown in the block diagram of the Contact Estimation in Figure 1, the equation of momental equilibrium then can be calculated as: (13)$$f_{c}=m\frac{g}{h}\Delta ZMP$$
where $f_{c}$ is the estimated contact force, *h* is the height of the CoM, *m* is the mass of the robot, g is the gravitational constant, and ΔZMP is the deviation of the zmp and the projection of gravity (PoG). The PoG can be calculated according to the kinematics analysis. The PoG is a fixed point when a robot is in a fixed configuration. In practice, the contact force calculated from Equation (Equation 13) is always non-zero. That means the robot will keep moving. We set a dynamic region for the PoG as the limits of the PoG. The contact is zero in this region. Otherwise it can be calculated according to Equation (Equation 13). The PoG limits then can be represented as an interval $\left[PoG-\mathrm{limits},PoG+\mathrm{limits}\right]$. The limits are as follows: (14)$$\mathrm{limits}=\left\{\begin{array}{cc} 0.1 & \frac{h}{r}\cdot \frac{a_{x}}{g}<0.1\\ 1 & \frac{h}{r}\cdot \frac{a_{x}}{g}>1\\ \frac{h}{r}\cdot \frac{a_{x}}{g} & \mathrm{else} \end{array}\right.$$

Thus, $feed_{1}$ and $feed_{2}$ add to the equation of the CPG model as: (15)$$\left\{\begin{array}{cc} feed_{1} & =-f_{c}\\ feed_{2} & =f_{c} \end{array}\right.$$

## 5. Results and Discussions

In this section, we implement several experiments to demonstrate the theoretical effectiveness. Our experiments are based on the simulation environment of the NAO biped robot. The simulation is executed on Webots software. The version of this software we use is R2022a. We program the controller using c++ programming language. (https://github.com/southwestCat/cpg-pulse.git (accessed on 18 January 2023)). The program can run on the Ubuntu 20.04 operating system with kernel version Linux 5.15 and gcc version 9.4.0.

### 5.1. Double-Leg Support

During the preparation stage of the experiment, the robot stands on the ground with double feet, and the height of the center of mass to the ground is set to 260 cm. When the biped robot is applied an increasing *x*-direction thrust, the output of the contact force estimation module will activate the compliant-resistant controller to let the robot’s CoM move forward to absorb the effect of the external force and then return back to restore balance.

As shown in Figure 5, the upper row shows a snapshot sequence of the NAO robot without CPG suffering a forward thrust. Figure 5(1) shows that the robot keeping balance initially stands on double feet and, at this time, applies a forward force to the centroid of it. Figure 5(2) shows that the center of mass of the robot moves forward under the action of external force, and Figure 5(3) shows that the centroid of the robot moves backward under the action of gravity. Figure 5(4) shows that the CoM of the robot moves back to a greater extent. Figure 5(5) shows that the robot returns to the equilibrium state through its own gravity without CPG. From the figures above that the center of mass of the robot will move backward with a large amplitude, resulting in a large deflection angle on the sagittal plane, perhaps causing the robot to fall.

In Figure 6a, the black line represents the contact force, calculated according to the ZMP measured value and the robot model, and the magenta line represents the CPG output equal to 0. At the beginning of the experiment, the robot stands on double feet and maintains still. The initial value of the ZMP is 20mm. An increasing *x*-direction thrust of 15N acts on the centroid of the robots at the moment of t=3s and lasts for 0.25s. As shown in Figure 6b, the black line represents the measured ZMP value, the red line represents the estimated value of PoG, and the green area represents the limits of PoG. From t=3s, the ZMP in the *x*-axis direction gradually increases in the positive direction. When t=3.25s, the ZMP value reaches the maximum and remains until 3.5s, and the maximum value is 70mm. After that, the ZMP value gradually decreases to 0 and continues to increase in the negative direction of the *x*-axis, and then falls again, back and forth, until the robot returns to the equilibrium state. When the curve of ZMP is inside the green area, it is considered that the robot is not under external force at this time. In Figure 6c, the blue line indicates the angle of the biped robot’s torso in the sagittal plane. At the moment of t=3s, the body angle increases forward around the *y*-axis, increases to the maximum, decreases to 0, and then increases in the opposite direction, back and forth. The robots restore balance in the end. When the external force acts on the robot, the trajectory of ZMP will exceed the PoG threshold, the robot’s CoM will move in the direction of force under the action of external force, and the body angle of the robot will rotate in the direction of the force. When the external force is withdrawn, the robot will swing back and forth under the action of robot gravity until it returns to equilibrium.

The lower row in Figure 5 shows a snapshot sequence of the NAO robot with CPG suffering a forward thrust. Figure 7 shows the variations of variables when the biped standing robot is under the action of the CPG model. In Figure 7a, the magenta line shows the output signal waveform of CPG outputs. An external force of 15N in the increasing *x*-axis at the moment of t=3s is applied to the robot and lasts for 0.25s. Figure 7c presents variations of the body angle around the *y*-axis.

Comparing Figure 7 with Figure 6, the robot with CPG signals will recover the unstable stage to the equilibrium state faster. Moreover, the rotation angle of the robot’s body is smaller, and it has experienced little body swing time. The output of CPG inhibits the robot from swinging backwards and forwards under the action of external force.

### 5.2. Single-Leg Support

During the preparation stage of the experiment, the robot stands on the ground with a single foot. The robot raises its right foot by 2cm and turns it into a left foot support, with the centroid offset by 50mm in the positive *y*-direction. When the biped robot is applied an increasing *y*-direction thrust, $F_{ext}=7\;N$ acting on the left shoulder of the robot and lasting for 0.5s, the output of the contact force estimation module will activate the compliant-resistant controller to let the robot’s hip joint rotate and the robot leans to the positive *y*-axis direction to absorb the effect of the external force, and then return back to restore balance.

The upper row in Figure 8 shows a snapshot sequence of the NAO robot without CPG suffering an increasing *y*-axis thrust. When a biped robot standing on a left foot is subjected to the force in the increasing *y*-direction, the robot moves in the positive direction first, and then the robot moves in the opposite direction. The robot’s right foot will land on the ground, and the robot will restore the support state of double feet. The lower row presents a snapshot sequence of the NAO robot with CPG. Under the modulation of the CPG signal, the robot first moves in the positive *y*-direction in the direction of external force, then moves in the negative *y*-direction. Finally, the robot restores balance and remains standing on its single left foot during the whole process.

From Figure 9 and Figure 10, we can see that when the biped robot stands on a single foot, it is difficult to restore its balance and still maintain its single-leg support through its gravity when subjected to external forces. However, the introduction of CPG into robot control effectively absorbs the effect of external forces, and the robot can recover its equilibrium state smoothly.

In summary, the compliant-resistant balance control based on Matsuoka’s CPG signals proposed in this paper can effectively restore the balance of the biped robot under the action of external forces, regardless of whether the robot is in the state of double-leg support or single-leg support. It is mentioned that the above experiments are only representative. When the robot is supported by two legs, it still has the same excellent performance when it is subjected to the force in the direction of the *y*-axis. When the robot is on single-leg support, whether it is lifting the right foot or the left foot, it can quickly and stably recover to a stable state. Simultaneously, it also has the ability to resist external forces in the *x*-direction.

## 6. Conclusions

The proposed balance-control method for biped robots uses a signal generator and modulator based on Matsuoka’s CPG model generating the defined pulse signals to achieve the compliant-resistant control for the biped robots. In this paper, the output of the CPG model is modulated to act on the CoM to generate a workspace trajectory or directly on the hip joint to generate a joint space trajectory. Under the control of the raised algorithm, the biped robot reduces body shaking and quickly returns to a balance state when subjected to external forces. The results prove that the biped robot control method based on the CPG model is a model-free control algorithm with a high research value.

The algorithm proposed in this paper, inspired by human’s response when suffering unknown perturbation, is a model-free balance-control method that overcomes the complexity of model-based control. We have demonstrated the validity of our proposed algorithm through simulation experiments. Whether the biped robot stands on single foot or double feet, the robot can restore balance regardless of the external force on the sagittal plane or the lateral plane. A signal generator we designed based on Matsuoka’ CPG model can generate defined signals. The balance control of a biped robot subjected to external force can be further studied regarding robot locomotion. 

## Figures and Tables

**Figure 1 biomimetics-08-00100-f001:**
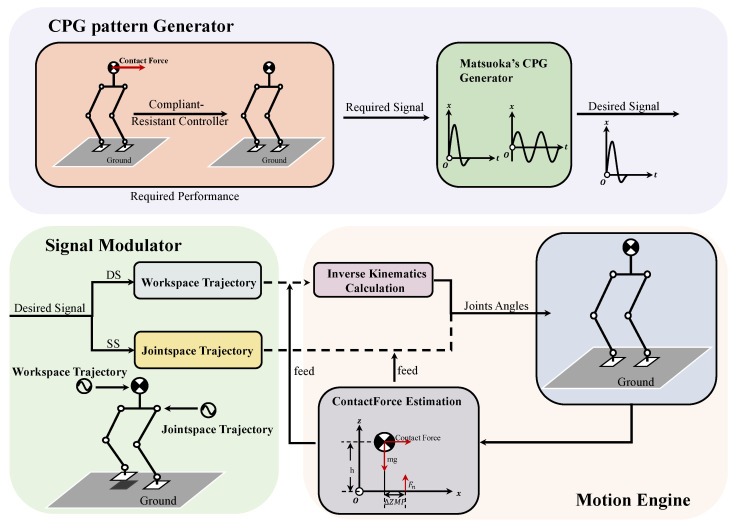
The architecture of the robot control system. In this diagram, the CPG pattern generator optimizes the output signal based on Matsuoka’s CPG model to the desired pulse pattern. The detailed generation and analysis of the signal are presented in Section 3. The signal modulator then modulates the desired signal to the workspace trajectory or the jointspace trajectory depending on the support types, double support or single support, and the deviation of the measured zmp and the projection of gravity is chosen as the feed term of the CPG model in Section 4. *h* denotes the height of the CoM, $F_{n}$ denotes the ground support force measured by the force sensors, *m* denotes the mass of the robot, g is the gravitational constant, and ΔZMP is the deviation of the measured zmp and the projection of gravity.

**Figure 2 biomimetics-08-00100-f002:**
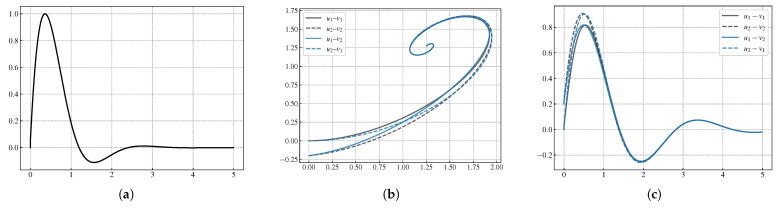
Generated Signal based on CPG by parameters $T_{u}=3.10412$, $T_{v}=0.424391$, $s_{0}=9.28835$, b=9.31659, $w_{fe}=-2.99902$. The signal decays to zero over time in the time domain. The phase portrait trajectory stays in one area as a whole denotes that the generated signal is stable. (**a**) The generated signal in the time domain. (**b**) Phase portrait of the generated signal. (**c**) Difference between state variables of the phase portrait.

**Figure 3 biomimetics-08-00100-f003:**
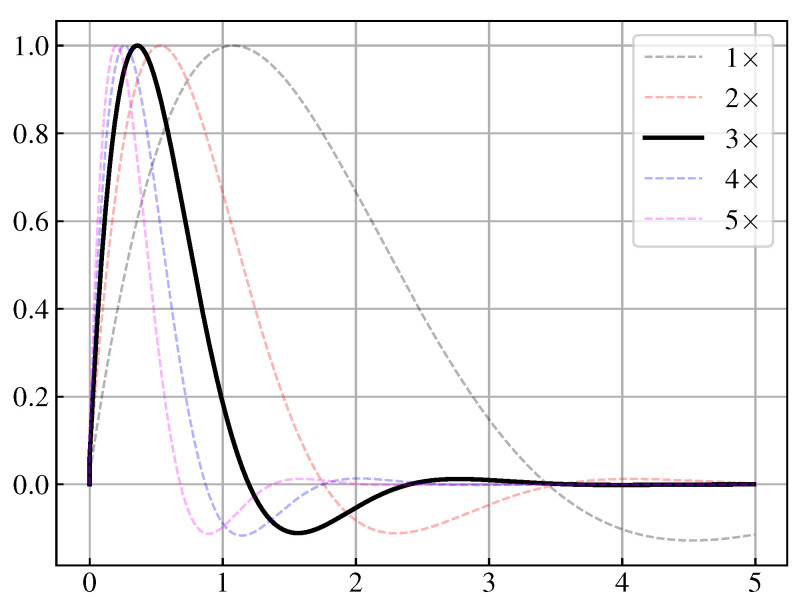
Generating Trajectories under different times of slope constraints.

**Figure 4 biomimetics-08-00100-f004:**
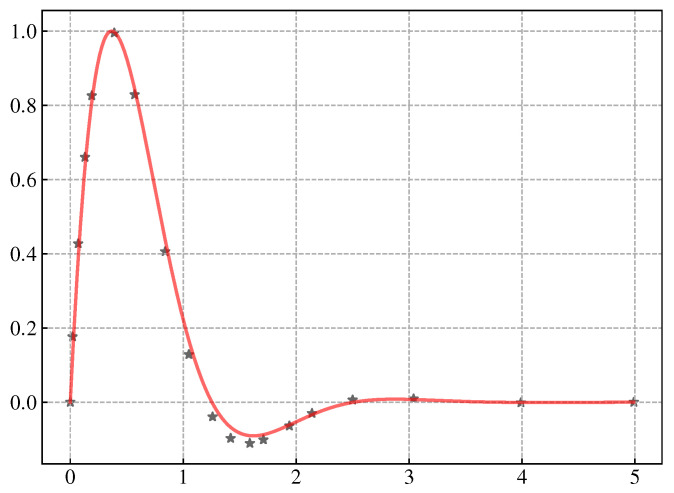
Template signal. Red line is the template function and the black star is the defined CPG output.

**Figure 5 biomimetics-08-00100-f005:**
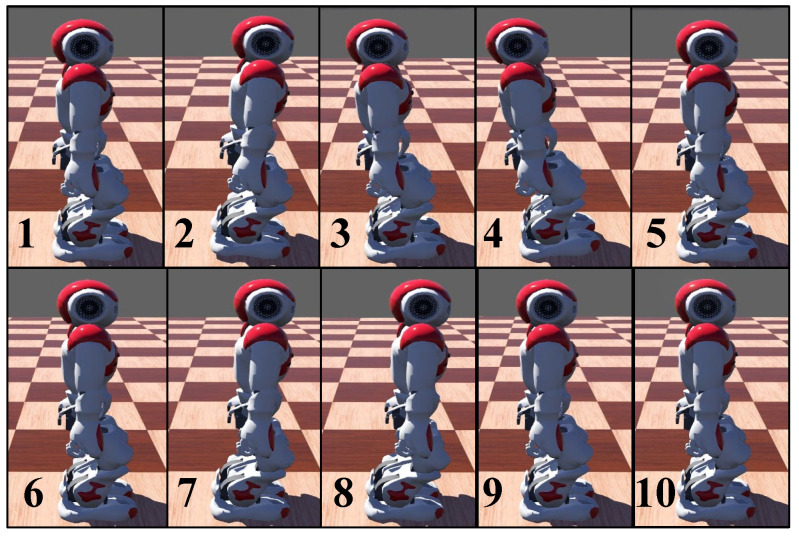
Snapshot sequences of double-leg support. The swing motion caused by the external force, $F_{ext}=15\;N$, is significantly suppressed under the action of the CPG controller. (**1**–**5**): without CPG, the robot swings forth and back under the external force. (**6**–**10**): with CPG, the robot moves its body forward to counteract the swing motion caused by the same external force and no backward movement.

**Figure 6 biomimetics-08-00100-f006:**
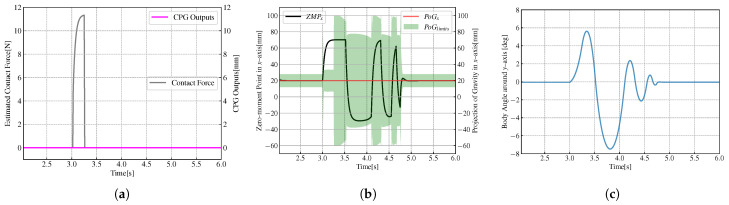
Variations of variables when the robots standing on double feet without CPG. (**a**) Variations of contact force and CPG output. (**b**) Variations of ZMP, the estimated projection of gravity (PoG) and the limit of PoG. (**c**) Variations of body angle.

**Figure 7 biomimetics-08-00100-f007:**
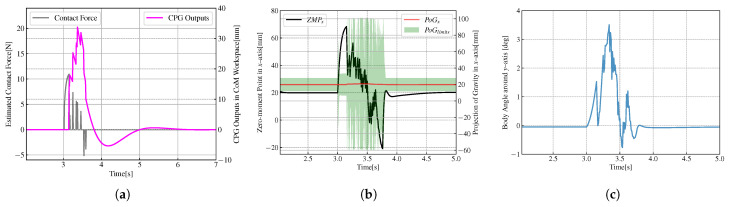
Variations of variables when the robots standing on double feet with CPG. (**a**) Variations of contact force and CPG output. (**b**) Variations of ZMP, the estimated projection of gravity (PoG) and the limit of PoG. (**c**) Variations of body angle.

**Figure 8 biomimetics-08-00100-f008:**
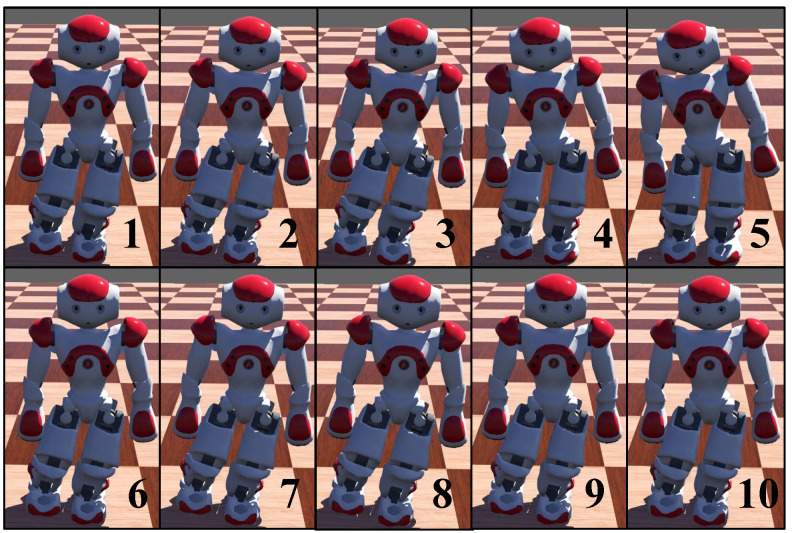
Snapshot sequences of single-leg support. The posture of standing on single support foot can be maintained even under the action of external force, $F_{ext}=7\;N$ acting on the left shoulder. (**1**–**5**): without CPG, the robot can not hold the posture, swing to left side first and then lean to the right side following the Newton’s first law of motion. (**6**–**10**): with CPG, the robot keeps standing on single support.

**Figure 9 biomimetics-08-00100-f009:**
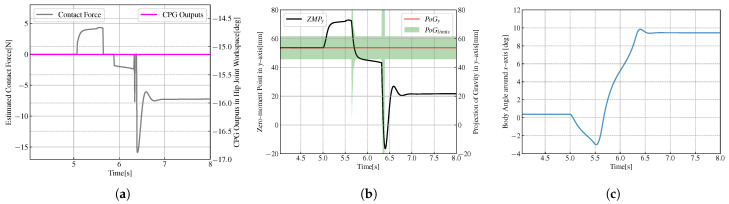
Variations of variables when the robots standing on a single foot without CPG. (**a**) Variations of contact force and CPG output. (**b**) Variations of ZMP, the estimated projection of gravity (PoG) and the limit of PoG. (**c**) Variations of body angle.

**Figure 10 biomimetics-08-00100-f010:**
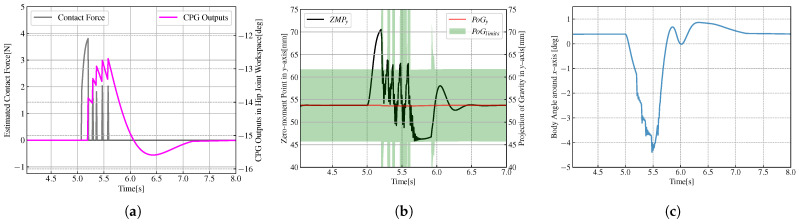
Variations of variables when the robots standing on a single foot with CPG. (**a**) Variations of contact force and CPG output. (**b**) Variations of ZMP, the estimated projection of gravity (PoG) and the limit of PoG. (**c**) Variations of body angle.

**Table 1 biomimetics-08-00100-t001:** The state variables in initial state and steady state.

$u_{1}$	$v_{1}$	$u_{2}$	$v_{2}$
0	0	0	−0.2
1.26932	1.26933	1.26932	1.26933

## Data Availability

Not applicable.

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
