# Peer review of "Central Pattern Generator with Defined Pulse Signals for Compliant-Resistant Control of Biped Robots"

_biomimetics, 2023, doi:10.3390/biomimetics8010100_

Round 1
Reviewer 1 Report
An article Central Pattern Generator with Defined Pulse Signals for Compliant-Resistant Control of Biped Robots is in subject of Biomimetics journal of MDPI
In article was proposed control strategy can generate defined pulse signals to realize compliant-resistant balance control for biped robots. It was checked by experiment using Nao robots.
This valuable paper need minor review for improve found mistakes
You can find my remarks in few comments for consider iin your review
Comment 1
In Introduction you wrote using numbering
These are the main contributions of this paper.
1. A CPG pattern generator is proposed based on the Matsuoka’s CPG unit. It generates
the desired pulse signals required to achieve the compliant-resistant balance control for the biped robots; 106
2. A signal modulator is raised to map the output signals of the CPG model to the
workspace trajectory (when standing on double feet) or joint space trajectory (when standing on a single foot) of the biped robots;
3. A novelty method is proposed to design Matsuoka’s CPG to generate the pulse signals required;
4. The proposed control strategy is validated in the biped robot NAO simulation environments.
Because numbering is used for section better is use of bullet
These are the main contributions of this paper.
· A CPG pattern generator is proposed based on the Matsuoka’s CPG unit. It generates the desired pulse signals required to achieve the compliant-resistant balance control for the biped robots;
· A signal mo……….
· …….
· The proposed control strategy is validated in the biped robot NAO simulation environments
Comment 2
You prepare article based on 25 references. Please consider to add more references in scientific subject of article
Sensors 2022, 22(12), 4440; https://doi.org/10.3390/s22124440
Current Robotics Report 2021, 2, 201–210; https://doi.org/10.1007/s43154-021-00050-9
Comment 3
You show
Figure 1. The architecture of the robot control system.
In Introduction
It need to move it to Section 2 after citation
Comment 4
You wrote
2. Control System Architecture
I propose use more detail name
2. Control System Architecture based on CPG
Comment 5
You present Section 3 as so large part of article which consist from 5 subsections
Please to consider add after
3. CPG-based Trajectory Generator
What you present in this section (based on titles of 5 subsections)
You wrote
The process of finding a parametric solution can be divided into the following steps:
1. Randomly generate an initial set of parameters;
2. Numerically integrate the CPG model under each parameter definition to obtain the time domain signal;
3. Evaluate the similarity of the obtained signal to the given signal characteristics, expressed in fitness;
4. Update parameters based on similarity;
5. Repeat the above process to get the parameter solution with the most prominent similarity.
I propose use of bullet for this part. And also for this part
In order to obtain a signal with compliant-resistant characteristics, we believe that the time-domain signal integrated by the CPG model should have the following characteristics:
1. The signal can complete the compliance and resistance action process within a given time range and remain stable;
2. The signal can eventually decay to zero;
3. The signal should have a particularly high energy, which is specifically reflected in the fact that the output amplitude of the original signal cannot be too small. Otherwise, the control signal will not reflect the characteristics of compliance;
4. The time integral of the absolute value of the signal cannot be too large. The CPG output signal corresponds to the control quantity of the robot controller, and the integral result corresponds to the cumulative value of the control quantity output by the controller. It’s desired that the controller can use as little control quantity as possible to achieve the control result.
Command 6
You wrote
A fitness function defined using signal correlation is expressed as:
fitness =correlation/autocorrelation (10)
Is it true?
Please consider to write
A fitness function defined using signal correlation and autocorrelation signals is expressed as:
Comment 7
You wrote Section 5
5. Results
I propose change name of this section on
5. Results and discussion
because you both present and discuss results of experiments
Reviewer 2 Report
Using a central pattern generator model based on Matsuoki’s model Xu et al investigate the ability of a biped robot to maintain balance when exposed to an external perturbation. The work is interesting and relevant. The manuscript is well-written. The figure caption could be improved. I have only minor comments, see below
Line 43: Suggesting an improvement of the sentence: “make humanoids not live up to the expectation of collaborating with humans and working in their environments. “ -> “make humanoids less suited for collaborating with humans and working in their environments.”
Line 64 and onward: “Most of the traditional CPG models….” Could the authors comment on the new CPG model which has rotational dynamics to explain rhythmic hindlimb movement? See Linden et al Nature 2022 (https://www.nature.com/articles/s41586-022-05293-w). Here they find experimentally, as opposed to the Matsuoka’s model, that there aren’t really “flexor” or “extensor” interneurons, but rather neurons representing a continuum of phases throughout the cycle, hence rotational population dynamics. It would be good if the authors could comment on this perspective and whether this has an impact on the viability of the model proposed here.
Figure caption: In general I recommend using more active language. The captions are all very short. Please help the reader more.
Figure 1- I dont think this figure is self-explanatory. Please explain what is going on in the figure.
Figure 2- the caption is very short. Perhaps more explanation would help the reader. “Generated signal” by what and for what? What is the conclusion of the figure? Why is it important?
Figure 5 and 8: is there any difference between the images and what is going on? What does this show or prove?
